# Pediatric Procedural Sedation and Analgesia (PROSA) in the Leuven University Hospitals: An Audit on Efficacy and Safety

**DOI:** 10.3390/children9060776

**Published:** 2022-05-25

**Authors:** Lotte Kerkhofs, Karel Allegaert, Jaan Toelen, Koen Vanhonsebrouck

**Affiliations:** 1Faculty of Medicine, KU Leuven, Herestraat 49, 3000 Leuven, Belgium; lotte.kerkhofs@student.kuleuven.be; 2Child and Youth Institute, KU Leuven, Herestraat 49, 3000 Leuven, Belgium; jaan.toelen@uzleuven.be; 3Department of Development and Regeneration, KU Leuven, 3000 Leuven, Belgium; 4Department of Pharmaceutical and Pharmacological Sciences, KU Leuven, 3000 Leuven, Belgium; 5Department of Clinical Pharmacy, Erasmus MC, 3000 GA Rotterdam, The Netherlands; 6Department of Pediatrics, University Hospitals UZ Leuven, 3000 Leuven, Belgium; koen.vanhonsebrouck@uzleuven.be

**Keywords:** pediatrics, procedural sedation and analgesia, anesthesia, non-pharmacological sedation

## Abstract

The hospital can be a stressful environment for a child. To address the increase in demands for pediatric procedural sedation, the PROSA team initiative was started in the Leuven University Hospitals in 2014. In this study, we assessed the efficacy and safety of this project since its initiation. Demographic (age, sex) and clinical (procedure, sedation method) data were prospectively registered by the dedicated PROSA team. Anonymized data (11/2014–6/2021) were extracted to a database for analysis. Data on 5090 procedures were available. The median age was 4.8 years. The sedation was successful in 98% of patients, be it classified as ‘technically difficult’ in 3.3%. Of the 2% of failed procedures, 69% were due to inadequate sedation and 31% to logistic reasons unrelated to sedation (such as puncture problem, suboptimal bowel preparation). The overall adverse event incidence was 2% and occurred predominantly during gastro-coloscopy or bronchoscopy. Pethidine/midazolam was used in 81 cases, nitrous oxide in 10 cases. A saturation decrease with oxygen supply was the documented adverse event in 83 cases. There were no deaths reported. With the current training and implementation, PROSA is effective with a success rate of 98% and an adverse event rate of 2%.

## 1. Introduction

A hospital is a setting where many procedures associated with pain and stress occur. Procedures such as venipuncture, wound dressing care or imaging can be painful or associated with stress, even more in children. Fortunately, efforts to reduce these burdens in hospitalized children have been developed and implemented. In the last couple of years, various tools were added to the range of procedural sedation techniques applicable in children. This makes it more feasible and easier to perform a range of medical interventions on children with a good tolerance and safety throughout the hospital and to tailor practices to their needs, however, this needs specific skills. Procedural sedation outside the operating theatre is hereby increasingly used, including sedation performed by non-anesthesiologists [1].

Current literature has a focus on the development and assessment of PPS (Pediatric Procedural Sedation) programs [2]. These are in-service training programs for healthcare providers to transfer and implement the knowledge needed to initiate and sustain PPS programs. Such programs are necessary to improve procedural sedation in children, related to both efficacy and safety. Studies evaluating these PPS programs hereby provide evidence on its usefulness and positive effects on various outcome measures, related to efficacy, safety, or patient satisfaction.

Along these lines, the University Hospitals in Leuven (UZ Leuven) have also developed and implemented a PPS program, called the Procedural Sedation and Analgesia (PROSA) project. PROSA is performed by a group of specially trained nurses who assist children when undergoing investigations or interventions in the hospital. These nurses hereby assess what type of tools can be applied to help children in the best possible way by making the procedure as atraumatic and comfortable as possible. This can be done through both non-pharmacological methods such as virtual reality glasses or the use of distraction, as well as pharmacological interventions such as nitrous oxide, lidocaine/prilocaine patches or dexmedetomidine.

The PROSA team treats the most anxious children and is involved in the more painful or stressful procedures. The project has prospectively collected data on these sedations since its initiation, but these data have not yet been audited, analyzed, or reported. This is important because the operational approach and structure of the PROSA project are unique in Belgium. In this audit, the efficacy and safety of the PROSA interventions will be described, while the dataset will be explored to identify potential trends in tools, patient numbers or characteristics over time, or procedure-specific findings on efficacy or safety, as a quality improvement audit.

## 2. Materials and Methods

This study is a retrospective analysis of prospectively registered data by the PROSA team from November 2014 to June 2021. It comprehends medical information registered at UZ Leuven in the context of the PROSA project. The information was systematically registered in a dedicated form by the nurse involved during and after each procedure within KWS (Clinical WorkStation), the medical software system of UZ Leuven. The head nurse of the PROSA team extracted the data from all these forms and anonymized the data by removing the patient’s name to ensure confidentiality. The structure of the forms changed twice. In 2019 and 2020, extra options were added for better registration for type of procedure and type of used sedative technique.

The data consists of different types of information: for example, date of sedation and date of birth. Based on this information, the age at the time of sedation was calculated. Other data extracted from the forms were: sex, American Society of Anesthesiologists (ASA) classification, unit where the procedure took place, physician and PROSA nurse involved, type of procedure needing sedation, type of sedative method used, and a free text space where the PROSA nurse could register special notes, problems with sedation or failure of procedure. Decisions on the various methods selected for a given child and a given procedure were made in consultation with patient, parents, and the PROSA team, including also the anesthesiologist. Microsoft Excel^®^ was used for arranging the data and for descriptive statistical analysis, such as mean, sum, and percentages.

After rearranging the full dataset for the 3 different consecutive types of forms, some other parameters had to be quantified. To do so, we used the free text space. We concluded that a procedure was successful when there was no mention of failed procedure. When it was written that the sedation had been difficult yet had a successful outcome, it was noted in an extra column. When notification of an adverse event was provided, it was included in the dataset. A distinction was made between failed procedures due to medical reasons or due to technical reasons. The free text spaces also included information about the tabulated information, such as product used, or type of procedure. When this was not already indicated in the previous columns, we adjusted the data for this extra information.

The pediatric department of UZ Leuven interacts closely with the PROSA team. PROSA provides services upon request by a healthcare provider from the pediatric department. They can be contacted by different routes. The first way in which the children’s hospital can consult the PROSA team is via the application form for specific procedures. In general, an application for PROSA is made for every child under the age of 4 for whom a CT scan or an MRI scan is planned. All bronchoscopies and procedures on endoscopy are planned with the assistance of the PROSA team, no matter the age. The second way, in the case of other types of procedures, the doctor or nurse can consult the PROSA team when difficulties related to pain or fear are expected. For example, when a venipuncture was stressful for the child a previous time, it will be noted to contact the PROSA team the next time. The third way to reach the PROSA team is via e-mail. Every child with an appointment in the children’s hospital will receive extra information about the functioning of the team and in what cases PROSA can assist. If parents think that this can be helpful for their child, they can also directly contact the PROSA team with their questions and request sedation.

The nurses of the PROSA team have received a dedicated/specific training to safely execute sedations. The education of the PROSA team consists of 4 different aspects. First, a PROSA nurse must have had at least 3 years of experience working as a pediatric nurse in the hospital setting. Second, these nurses follow a two-day course on how to approach children, to understand pain in children, and the different products used by PROSA (administration, pharmacokinetics, contra-indications). Third, the nurse follows an internship with the PROSA team for 2 days. Fourth, they take a yearly class on Intermediate Pediatric Life Support, as first responders in the event of problems during the sedation. This level of training has to be understood as part of the broader framework, including the availability of a pediatric resuscitation team and the fact that the anesthesiologist is pre-informed. Every sedation is performed under supervision of anesthesiology. Patients are being screened carefully by an anesthesiologist for possible contra-indications for sedation. For example, when dexmedetomidine will be administered during the procedure, the possible candidate for sedation will be screened for pulmonary or cardiac abnormalities. Most patients with ASA 3 or 4 are excluded for sedation by PROSA. On the day of sedation, PROSA will inform the anesthesiology department when and where the sedation will occur, which is an additional safety precaution.

Fifth, there is a yearly mandatory training in the operating theatre where the main goal is to learn how to operate a balloon mask. This is an important skill because the PROSA nurse has to bridge the time between onset of acute event and arrival of the physician. This day is organized in the children’s operating theater with the supervision of anesthesiologists. They hereby also learn how to place a Mayo canula.

## 3. Results

### 3.1. Data Collection

A total of 5475 sedations were collected. Two exclusion criteria were subsequently applied (see Figure 1). The first exclusion criterion was an upper limit of age. The PROSA team performs sedation on children, but rarely on patients older than 18. As this study focuses on pediatric sedation, 69 sedations were excluded because of the patient being older than 18. The second exclusion criterion were premature babies and neonates admitted to the neonatal (intensive) care unit. All patients hospitalized in the units of neonatal intensive care were excluded from data analysis. This came down to 298 sedations. After adjusting for these exclusion criteria, the dataset consisted of 5108 sedations. While arranging the database, 12 sedations were registered twice and 6 sedation registrations were administrative (i.e., advice via a phone call). The final database consisted of 5090 sedations.

### 3.2. Demographic Data

Demographic data are listed in Table 1.

In Table 2, a distribution according to age groups was established, according to the National Institute of Child Health and Human Development (NICHD) Pediatric Terminology [3].

### 3.3. Timeline of Sedations

The PROSA project started in November 2014. The first registered sedation dates from 28 November, the start of the PROSA project. The latest sedation included in the database was on 14 June 2021. The distribution of the sedations according to year can be found in Table 3. Only the first half of the year 2021 was included in the data, explaining the lower figure. 

Figure 2 is a representation of the number of sedations per month. 

### 3.4. Procedures Assisted by PROSA

A total of 5350 procedures were registered in the PROSA database. This number is higher than the number of sedations (*n* = 5090), because sometimes several procedures were performed during the same sedation. The procedures needing the support of the PROSA team are listed in Table 4.

When all nuclear diagnostic procedures were pooled, we reached 17.7%. For these procedures (MIBG, bone scan, and PET scan together with EC scan and DMSA scan) the same type of sedation (dexmedetomidine) was used. The table shows the full range of different procedures. When a sedation was registered but no procedure was indicated, we counted this as one unknown procedure.

The category ‘other’ comprehends an extended range of procedures. A non-exhaustive list from the more commonly reported to least common reported procedures is provided: urinary catheter placement, DEXA scan, cystography, removing molluscum contagiosum, otolaryngology examination, Meckel’s scan, stitching wound, muscle biopsy, RX colon, botulinum toxin injection, kidney biopsy, ophthalmology examination, or proton therapy.

There are 1802 procedures in the category gastro-coloscopy. This is the sum of both ‘gastro/coloscopy’ (1675 procedures) and ‘coloscopy’ (127 procedures). In 31 sedations, these 2 procedures were simultaneously registered. We suspected the distribution to be 31 combination procedures, 96 coloscopies, and 1644 gastroscopies. The head nurse of the PROSA team (K.V.) confirmed that sedation of the combination of these 2 procedures was not frequently assisted by PROSA. Most procedures on endoscopy are gastroscopies, as the numbers show. For these 3 types of procedures, 1 sedation strategy has been used: pethidine (1 mg/kg with a maximum of 50 mg) + midazolam (0.1 mg/kg with a maximum of 5 mg).

For the 485 MRI scans, 170 were in the context of the COSMO project [4]. Initially, the COSMO protocol consisted of telling the story of a shooting star. This protocol has been transformed and developed into an app for children to playfully prepare for upcoming MRI scans. As children must lay still for up to 20 min in a surrounding with sudden loud noises and unfamiliar narrow space, the make-believe is that they are traveling to space. The goal is to increase the number of pediatric MRI scans without the need of pharmacologically-induced sedation, so that it is an integrated part of the PROSA program.

### 3.5. Sedative Methods Applied

Many different tools can be used as a method of sedation or analgesia. The most frequently used sedative techniques were a vacuum mattress, very commonly combined with distraction, nitrous oxide, and dexmedetomidine. Other products are listed in Table 5.

PROSA uses different types of non-pharmacological methods for sedation. A MedVac Mattress^®^ (Kohlbrat & Bunz, Radstadt, Austria) is a vacuum immobilization mattress with a multifunctional purpose. PROSA uses it in combination with a sedative (dexmedetomidine or chloral hydrate) for imaging in children. It helps with comforting the child who is already sedated and making sure high-quality imaging can be performed. It keeps the child in a certain position during the sedation, as well as having the additional functional aspect of the “cocoon effect” that makes the children sleep longer. They can also be transferred easily from their bed to the imaging table. Especially in very young children, the mattress is frequently used. 

Another non-pharmacological way of comforting children is by distraction. By talking, interacting, and playing with the child, the nurse tries to distract the child from the stressful environment. Distraction also includes the use of virtual reality glasses. This is a relatively new asset to the product range of procedural sedation. 

Buzzy^®^ (Pain Care Labs Buzzy XL Healthcare, Atlanta, GA, USA) is a cooling icepack with a vibrating motor that is placed on the location of the needle puncture, for example, before venipuncture [5]. It is called ‘buzzy’ because it is shaped like a bee. It is a promising tool in the PPS care. Glucose 30% in an oral solution is used in babies undergoing medical imaging. It soothes babies and has a short analgesic effect. In newborns, this is as effective for imaging as the sedative product midazolam and has no side effects [6].

The PROSA nurses have the authorization to administer some type of sedative medication to children, following prescription by the treating physician. The sedation is carried out under supervision of anesthesiology. In UZ Leuven, a fixed nitrous oxide mixture (50% oxygen and 50% nitrous oxide) is used, as this has an anxiolytic effect when inhaled [7]. Nitrous oxide in monotherapy can be administered by all nurses of UZ Leuven. The PROSA team uses it in combination with other medication such as fentanyl (intranasal) or midazolam in specific settings, such as skin or muscle biopsy.

Before December 2017, chloral hydrate was frequently used. Dexmedetomidine was added to the range of tools for procedural sedation in December 2017. It is used in children older than 4 weeks and can replace chloral hydrate in this age category. Dexmedetomidine has a shorter half-life than chloral hydrate, and it has less effects on the respiratory and cardiac system [8]. Chloral hydrate is now only used in children younger than 4 weeks old because of a lack of evidence of safe use of dexmedetomidine in this age category. 

The combination of midazolam and pethidine is used during endoscopy. Procedures on endoscopy are a large part of PROSA’s daily tasks. The number in the Table 4 is an underestimation of the exact number, due to less accurate registration. The timing of these 2 products is very strict. Pethidine must be injected 15 min before the start of a procedure, while midazolam has an immediate effect when injected.

In UZ Leuven, two types of topical anesthetics are used: Emla^®^ (Aspen Pharmacare Australia, St. Leonards, Australia) and Rapydan^®^ (Eurocept Pharmaceuticals, Ankeveen, The Netherlands) [9]. Emla^®^ is a lidocaine (25 mg/1 g) and prilocaine (25 mg/1 g) is a cream, and Rapydan^®^ is a lidocaine (25 mg/1 g) and tetracaine (25 mg/1 g) is a cream. Emla^®^ takes 60 min for optimal functioning, where Rapydan^®^ needs 30 min. 

When administration route was indicated, we collected this information in the second part of Table 5. The intravenous route of administration (midazolam, pethidine) accounts for more than one third of all performed sedation. This number is so high because it is related to the number of procedures on endoscopy. During a gastro- or coloscopy, both pethidine and midazolam are administered via intravenous access. Dexmedotomidine (imaging) and fentanyl (skin or muscle biopsy) are restricted to intranasal administration for the PROSA nurses. 

### 3.6. Efficacy

Table 6 shows the data on efficacy of the PROSA team. A successful procedure was stated in 4990 patients. When sedation was difficult (*n* = 169), but the procedure could still be carried out, we classified this in a subcategory of the successful procedures.

We divided the group of failed procedures into two main categories. The first category is failed procedure due to inadequate sedation. The table gives an indication of the 5 most frequent procedures and the 5 most frequent tools used for sedation where an adverse event occurred. In the second column, a comparison is made with the total amount of that respective type of procedure or used tool for sedation. In 4.8% of blood sampling, sedation was not effective. A frequently used method of analgesia in this procedure is the Emla^®^ or Rapydan^®^ cream. As stated in the table, the same number can be retrieved when looking at the failed sedation rate of topical anesthesia. Distraction was used in 28 cases of failed procedures. In 23 of these cases, another product besides distraction was used. In only 5 cases, the used tool was distraction alone.

The second category of reason for a failed procedure is unrelated to sedation itself but comes from logistic reasons. This accounts for 31% of failed procedures. The different reasons can be seen in the table. ‘Other’ comprehends a collection of different problems, listed from most to least common: planning error for CT scan, communication problem (sedation had to be performed by an anesthesiologist), divergent anatomy, and not fasting before a CT scan.

As seen in Table 7, the distribution of failed procedure by age is equally spread. There is no age group where sedation by PROSA is less efficient.

Table 8 shows the failed sedation rate over the course of years. We do not observe an increase or decrease in the number of failed procedures. The overall rate of failed sedation is very low. The percentages in the table are the ratio of failed sedations to the total number of sedations in the respective year.

### 3.7. Safety

The overall adverse event incidence was 2.02% (103 cases) (see Table 9). Desaturation with the need of oxygen therapy was the most commonly observed adverse event. In some cases, oxygen was initiated to control oxygen saturation. Neurologic problems were headache or dizziness after the end of procedure, and in one case, trembling, and an episode of epilepsy. Cardiac problems were episodes of bradycardia. ‘Other’ comprehends an extended range of adverse events listed: petechia, rash or swollen eyes after pethidine, swollen eyes after nitrous oxide, a sore throat, coughing, and epistaxis. There were no deaths or permanent damage reported.

Adverse events occurred predominantly during gastro-coloscopy and bronchoscopy (see Table 10). The other adverse events occurred during medical imaging. During these procedures, more frequently, medication with a systemic effect was used, for example, the combination of pethidine and midazolam.

Table 11 shows the adverse events according to type of sedative method used. The high ratio of adverse events on the total number of sedations using a combination of midazolam and pethidine can be explained by the low number of registrations of this type of sedation, as stated earlier.

Table 12 shows the distribution of adverse events according to age. When looking at the percentage of adverse events in each age group, an equal distribution is seen.

In Table 13, the evolution of adverse events in time can be seen. A decrease in the number of adverse events can be noted in the year 2020 and 2021. Most of the adverse events happened during bronchoscopy and endoscopic procedures. In October 2019, the anesthesiology department took over the sedations during bronchoscopy. In May 2020, the protocol of the sedation during endoscopy was adjusted with a more optimized use of pethidine. In October 2021, the anesthesiology department took care of these sedations as well, and they can use more effective sedatives such as propofol that result in less adverse events. 

## 4. Discussion

This audit analyzed the PROSA database on efficacy and safety since its initiation in UZ Leuven. Our findings are in line with other reports, indicating that PPS programs performed by a trained sedation team are highly effective and have a low rate of adverse events. Previous studies have shown similar results. A systematic review from 2016, including 41 articles, has evaluated the incidence of adverse events during PPS by physicians in the emergency department [10]. In 1.5% of 11,457 sedations, hypoxia occurred. In a study in 2003, 17.9% adverse events were found with the use of midazolam in combination with another product by non-anesthesiologists [11]. This can be compared to our numbers of adverse events during broncho- or endoscopy, where midazolam was also used in combination with other products (pethidine).

The low adverse event rate must be sustained. This is achieved by the above mentioned safety rules and structured approach (prescreening, structured contact before initiation of pharmacological sedation, structured training) Most patients with ASA 3 or 4 were excluded for sedation by PROSA, because sedation in these categories is prone to more adverse events [12]. According to a European survey in 2021 of senior physicians, a lack of protocol and training is the main reason for dissatisfaction with management of anxiety and pain in children in their hospital setting [13]. They also found that only a third of the sites had the complete staff certified in a Pediatric Advanced Life Support course. In a Dutch survey conducted in 2017, the importance of improving PPS training, as well as increasing emergency physician staffing was highlighted [14].

Basic principles on how to perform sedation and how to comfort children should be known by all nurses on the emergency and pediatric department. They can perform sedation with, among others, distraction, virtual reality glasses, topical anesthetics, or nitrous oxide in monotherapy. When a combination of different products is needed, PROSA can be contacted to assist. Some products, such as midazolam, dexmedetomidine or pethidine, must be administered exclusively by PROSA or anesthesiology. This is an agreement within UZ Leuven.

It is relevant to stress that we also considered failed sedation as a relevant adverse event. Although about 30% related to logistics, there is still a very small group where the intended approach failed. In these cases, a pragmatic approach was taken considering the available resources (e.g., availability of anesthesia team) to proceed as a step up approach. Alternatively, the procedure had to be rescheduled. Although we are aware that this is not perfect for the individual child and family, it is a reasonable balance between patient care and resource allocation. Along the same lines, higher failure rates for specific procedures (Table 10) resulted in a shift from PROSA to anesthesia for gastro-coloscopy or bronchoscopy. The PROSA team were not to substitute general anesthesia as needed, but rather, they covered additional needs for sedation (Table 4).

As illustrated in Figure 2, an overall increase in the monthly number of sedations can be seen, including two dips in the curve. The first dip can be found at the end of the first working year of the PROSA team when the healthcare workers were still getting acquainted with the project. Additionally, at the start of the project, the team was not completely staffed so continuity could not be guaranteed as there was not always a PROSA nurse available. This might have discouraged medical professionals to call and rely on PROSA assistance at the start of the project. There is a second dip mid-2020 that coincides with the start of the COVID-19 pandemic. At that time in Belgium, healthcare functioned on a lower level with a limitation on surgeries and invasive procedures. This is also the reason for the lower number of sedations at the end of the curve.

The increase in activities of the PROSA team is partly because its applications are better known in the hospital, and parents of young patients are starting to rely on its advantages. Its growth is also due to more innovative techniques, such as dexmedetomidine, which gives PROSA the opportunity to help more children with less aggressive sedation techniques.

## 5. Conclusions

With the current training and implementation of the PROSA team, PPS is effective with a success rate of 98.0% and a low adverse event rate of 2%, and rather clustered in gastro, colo- or bronchoscopy. No deaths or severe events were reported. The program induced significant changes in practices and is widely accepted in our institution. 

## Figures and Tables

**Figure 1 children-09-00776-f001:**
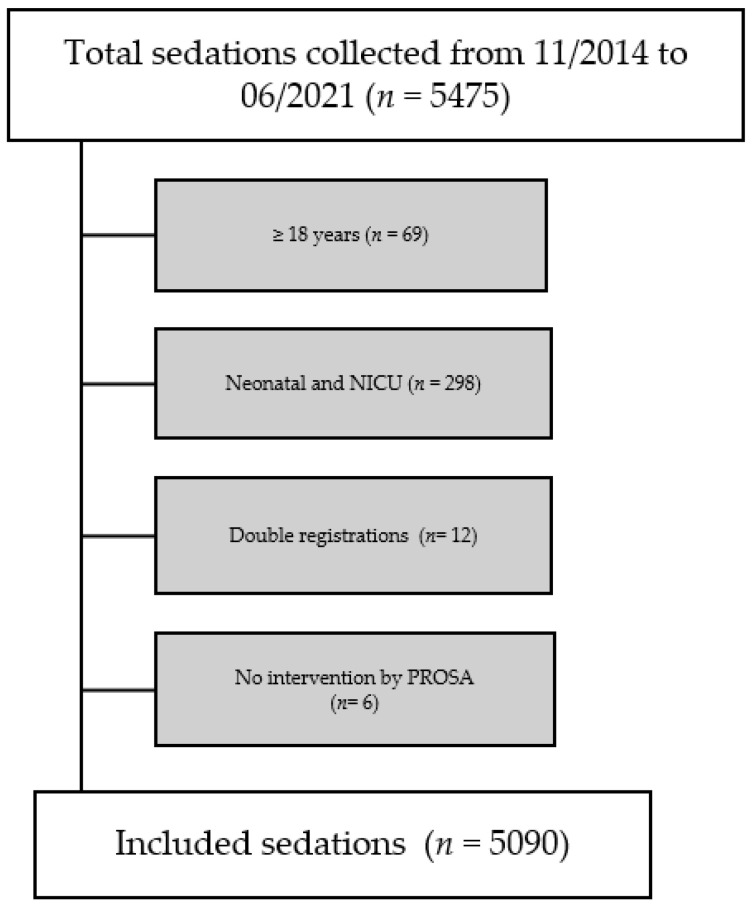
Flowchart of collected data and patient inclusion. Abbreviations in the figure: NICU = Neonatal Intensive Care Unit, PROSA = Procedural Sedation and Analgesia.

**Figure 2 children-09-00776-f002:**
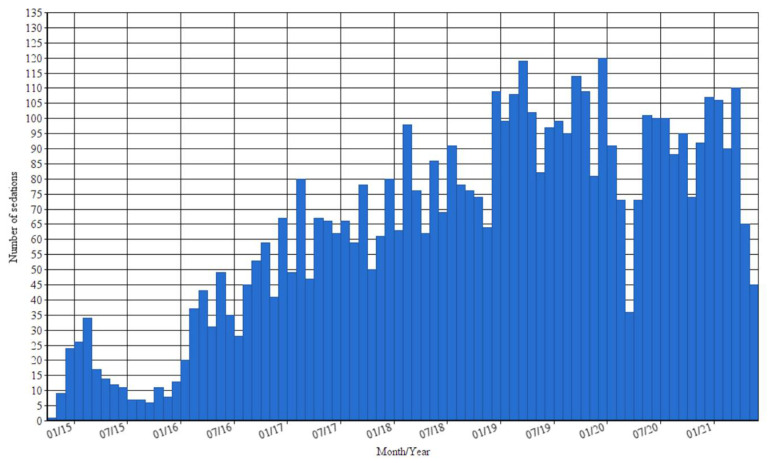
Number of sedations per month from November 2014 to June 2021.

**Table 1 children-09-00776-t001:** Summary of the demographics of patients receiving pediatric procedural sedation.

Variable	Level	*n* (%) (*N* = 5090)
Age (years)	Median	4.8
	25–75%	1.73–10.72
	Range	0.02–17.99
Sex	Male	2701 (53.06%)
	Female	2389 (46.94%)
ASA classification	I	3503 (68.82%)
	II	1270 (24.95%)
	III	240 (4.72%)
	IV	6 (0.12%)
	Unknown	71 (1.39%)

Abbreviations in the table: ASA classification = American Society of Anesthesiology classification.

**Table 2 children-09-00776-t002:** Age groups (Stages defined according to NICHD Pediatric Terminology).

Stage	Age	*n* (%) (*N* = 5090)
Neonatal	0–27 days	48 (0.94%)
Infancy	28 days–12 months	776 (15.25%)
Toddler	13 months–2 years	591 (11.61%)
Early childhood	2–5 years	1482 (29.12%)
Middle childhood	6–11 years	1179 (23.16%)
Early adolescence	12–18 years	1014 (19.92%)

Abbreviations in the table: NICHD = National Institute of Child Health and Human development.

**Table 3 children-09-00776-t003:** Number of sedations per year from initiation onwards.

Year	Number of Procedures
2014	10
2015	177
2016	454
2017	752
2018	917
2019	1214
2020	1043
2021	523

**Table 4 children-09-00776-t004:** Summary of the 5350 procedures assisted by PROSA.

Type of Procedure	*n* (%)
Gastro-coloscopy	1802 (33.68%)
DMSA scan or EC scan	722 (13.50%)
CT scan	535 (10.00%)
MRI scan	485 (9.07%)
Bronchoscopy	256 (4.79%)
Venipuncture	228 (4.26%)
Lumbar puncture	206 (3.85%)
IV catheter	139 (2.60%)
Bone scan	89 (1.66%)
MIBG	76 (1.42%)
IM injection	61 (1.14%)
PET scan	60 (1.12%)
Nose mucosal biopsy	46 (0.86%)
Wound care	46 (0.86%)
PICC procedure	43 (0.80%)
PAC puncture	42 (0.79%)
Nusinersen IT injection	41 (0.77%)
Applying cast	38 (0.71%)
Urology examination	38 (0.71%)
Nasogastric tube	34 (0.64%)
Unknown	39 (0.73%)
Other	339 (6.34%)

Abbreviations in the table: DMSA = Dimercaptosuccinic Acid, EC = Ethylene dicysteine (renal scan), CT = Computed Tomography, MRI = Magnetic Resonance Imaging, IV = Intravenous, MIBG = Meta-iodo-Benzyl-Guanidine, IM = Intramuscular, PET = Positron Emission Tomography, PICC = Peripherally Inserted Central Catheter, PAC = Port- A- Cath, IT = Intrathecal.

**Table 5 children-09-00776-t005:** Summary of methods of sedation applied in 5090 patients.

Type of Product	*n* (%)	Administration Route		*n* (%)
MedVac Mattress	1098 (21.57%)	IV		1832 (35.99%)
Mattress alone	46 (0.90%)	Nasal		696 (13.67%)
Distraction	1086 (21.34%)	Rectal		478 (9.39%)
Distraction alone	333 (6.54%)	Oral		222 (4.36%)
Distraction + mattress	64 (1.26%)			
Nitrous oxide	791 (15.54%)			
Dexmedetomidine	502 (9.86%)			
Midazolam-Pethidine	301 (5.91%)			
Topical anesthetic	266 (5.23%)			
Glucose, oral	154 (3.03%)			
Chloral hydrate	147 (2.89%)			
Midazolam	88 (1.73%)			
Fentanyl	16 (0.31%)			
Buzzy^®^	10 (0.20%)			
Unknown	146 (2.87%)			

Abbreviations in the table: IV = Intravenous.

**Table 6 children-09-00776-t006:** Overall efficacy of sedations and failed procedure according to used method and type of procedure.

Total	*n* (%) (*N* = 5090)	*n* (%) (*N* = Total in Respective Subcategory)
** *Successful procedure* **	** *4990 (98.04%)* **	
Difficult sedation yet successful procedure	169 (3.32%)	
** *Failed procedures* **	** *100 (1.96%)* **	
**Inadequate sedation**	**69 (1.36%)**	
Venipuncture	11 (0.22%)	4.82% of 228
CT scan	7 (0.14%)	1.31% of 535
Gastro-coloscopy	7 (0.14%)	0.39% of 1802
Lumbar puncture	7 (0.14%)	3.40% of 206
MRI scan	5 (0.10%)	1.03% of 485
Distraction	28 (0.55%)	2.58% of 1086
Nitrous oxide	26 (0.51%)	3.29% of 791
Topical anesthetic	13 (0.26%)	4.89% of 266
Dexmedetomidine	8 (0.16%)	1.59% of 502
Distraction alone	5 (0.10%)	1.50% of 333
**Circumstantial reason**	**31 (0.59%)**	
Puncture/access problem	7 (0.14%)	
Food in stomach/bowel during endoscopy	7 (0.14%)	
Failed procedure not related to sedation	5 (0.10%)	
Extravasation	4 (0.08%)	
Other	8 (0.16%)	

Abbreviations in the table: CT = Computed Tomography, MRI = Magnetic Resonance Imaging.

**Table 7 children-09-00776-t007:** Failed procedure in terms of age of the child.

Age	*n* (%) (*N* = 5090)	% of Specific Age Category
0–27 days	0 (0%)	0% of 48
28 days–12 months	10 (0.20%)	1.29% of 776
13 months–2 years	7 (0.14%)	1.18% of 591
2–5 years	30 (0.59%)	2.02% of 1482
6–11 years	27 (0.53%)	2.29% of 1179
12–18 years	26 (0.51%)	2.56% of 1014

**Table 8 children-09-00776-t008:** Failed procedure according to year from initiation onwards.

Years	*n* (%)
2014	1 (10.00%)
2015	8 (4.52%)
2016	6 (1.32%)
2017	14 (1.86%)
2018	18 (1.96%)
2019	26 (2.14%)
2020	14 (1.34%)
2021	13 (2.49%)

**Table 9 children-09-00776-t009:** Type of adverse event which occurred during sedation by PROSA.

Total	*n* (%) (*N* = 5090)
Adverse events (cases)	103 (2.02%)
Desaturation, oxygen administered	83 (1.63%)
Neurologic problem	7 (0.13%)
Cardiac problem	5 (0.01%)
Nausea	4 (0.01%)
Death	0 (0%)
Other	9 (0.18%)

**Table 10 children-09-00776-t010:** Occurrence of adverse event according to type of procedure (one procedure could result in several adverse events (Table 9).

Type of Procedure	*n* (%) (*N* = Total Number of the Respective Procedure)
Gastro-coloscopy	45 (2.69%)
Bronchoscopy	40 (15.63%)
MRI scan	3 (0.62%)
CT scan	3 (0.56%)
Bone scan	1 (1.12%)
DMSA scan	1 (0.18%)
EC scan	1 (0.62%)
PET scan	3 (5%)

Abbreviations in the table: MRI = Magnetic Resonance Imaging, CT = Computed Tomography, DMSA = Dimercaptosuccinic Acid, EC = Ethylene dicysteine (renal scan), PET = Positron Emission Tomography.

**Table 11 children-09-00776-t011:** Occurrence of adverse event according to method of sedation used.

Type of Product	*n* (%) (*N* = Total Number of Respective Method of Sedation)
Midazolam-Pethidine	81 (26.91%)
Nitrous oxide	10 (1.26%)
Chloral hydrate	4 (2.72%)
Dexmedetomidine	2 (0.40%)
Topical anesthetic	1 (0.38%)
Glucose, oral	3 (1.95%)
Midazolam	2 (2.27%)

**Table 12 children-09-00776-t012:** Occurrence of adverse events according to age of the child.

Age	*n* (%) (*N* = Total Number of Children in Respective Age Category)
0–27 days	0 (0%)
28 days–12 months	21 (2.71%)
13 months–2 years	9 (1.52%)
2–5 years	34 (2.29%)
6–11 years	24 (2.04%)
12–18 years	15 (1.48%)

**Table 13 children-09-00776-t013:** Occurrence of adverse events per year from initiation onwards.

Years	*n* (%) (*N* = Total Number of Sedations in Respective Year)
2014	2 (10%)
2015	2 (1.13%)
2016	3 (0.66%)
2017	26 (3.46%)
2018	27 (2.94%)
2019	30 (2.47%)
2020	10 (0.96%)
2021	3 (0.57%)

## Data Availability

The corresponding author will welcome a request for data sharing or access, based on a reasonable request and if it is supported by study protocol.

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
