# Peer review of "Pediatric Procedural Sedation and Analgesia (PROSA) in the Leuven University Hospitals: An Audit on Efficacy and Safety"

_children, 2022, doi:10.3390/children9060776_

Round 1

Reviewer 1 Report

Thank you for this nice audit.

I would like to know if, in case of difficult or inadequate sedation, another level of sedation was planned in advance. Are step up anticipated?

16 patients received Fentanyl, how do you establish the limit between sedation and anesthesia?

In term of quality of the procedure and patient/parent satisfaction for endoscopy and bronchoscopy, is it comparable to general anesthesia?

PROSA seems safe and effective but is it as safe and effective as anesthesia for more invasive procedure?

Author Response

Thank you for this nice audit.

We thank the reviewer for the very supportive overall assessment of the paper.

I would like to know if, in case of difficult or inadequate sedation, another level of sedation was planned in advance. Are step up anticipated?

Step-up approaches are not preplanned, but are obviously possible. However, this necessitates the availability of a team with higher skills so that this was decided on an ad hoc basis, either shortly following a failed sedation, or more likely, after reshedulement. In this way, we can find a reasonable balance between resource allocation and the needs of individual patients.

The plan is always discussed with the patient and the parent and we make a realistic plan according to our possibilities, the previous experiences of the patient and, if needed, the advice of anesthesia. Sometimes the procedure is planned straight away with anesthesia if we anticipate for a difficult sedation or the need of deep sedation. In case of difficult of inadequate sedation the procedure is rescheduled and planned with anesthesia. This has been added to the paper.

16 patients received Fentanyl, how do you establish the limit between sedation and anesthesia?

The use of Fentanyl by Prosa-nurse is restricted in 2 ways. When preagreed, nurses are only allowed to give Fentanyl Intranasally and the use is pre-defined on a small number of procedures. (for example : biopsy of the skin of muscle combined with Nitrous Oxyde) If we anticipate or observe that the analgesia prescribed is not enough, we call anesthesia for advice. If deeper sedation is needed, the patient is done by anesthesia.

In term of quality of the procedure and patient/parent satisfaction for endoscopy and bronchoscopy, is it comparable to general anesthesia?

As the data showed, it turned out that the relative risk of failure was somewhat higher (cfr also comment of reviewer 2) for these procedures, at least suggesting that this was not similar. Based on these findings, we have in the meanwhile adapted the approach taken, and provide general anesthesia as first approach for these procedures. Our experience as Prosa-nurse was, that especially the youngest patients, weren’t deep enough sedated for a comfortable procedure. There was certainly room for improvement on that aspect. In the literature, it is also stated that general anesthesia is safer and more comfortable than sedation. We assume that this is improved especially with the range of medication anesthesia has. We have added this to the revised version of the paper.

PROSA seems safe and effective but is it as safe and effective as anesthesia for more invasive procedure?

Likely not, and the PROSA approach is not implemented instead of general anesthesia, but rather to cover another need (cfr procedures described). We  would even state that for more invasive procedure, as Prosa-nurses, we do not have the correct tools (restriction of use of medication by Belgian law) to work as effective as anesthesia. That’s why we work very closely with anesthesia to take the appropriate decision on whether prosa would do the sedation on certain patients. Being aware of the limitations and restrictions is crucial. This reflection has been added in the discussion section of the paper.

Reviewer 2 Report

The authors retrospectively look at their Pediatric Procedural Sedation and Analgesia program with an eye towards safety and efficacy. I applaud the authors on taking this important step in program evaluation to continuously improve the program. The manuscript would benefit from consideration of the points listed below. 

Introduction: overall written colloquially. It could benefit from a review

Materials and Methods

  1. Authors should better define how various methods were chosen by the PROSA team. Where there guidelines or were they "prescribed" by the anesthesiologist overseeing the team?
  2. Line 112: a couple of years is vague. The authors should better define the criteria for experience 
  3. Line 117: why was only intermediate pediatric life support required? The authors reference in line 345 that a survey found that only a third of sites were advanced life support certified which assumes that this is the standard

Results:

  1. For those MRI scans done in the context of the COSMO project, was this administered and trained by the PROSA team? If not, this may not represent an accurate assessment of the PROSA program for MRI. 
  2. Table 5: is there a difference between MedVac Mattress and Mattress alone? The first line on many of the tables are underlined and it is unclear why or if there is an intentional separation 
  3. Table 5: Which medications were administered using those administrative routes? For example some medications have multiple routes of administration. There is a brief but incomplete expiation in the paragraph starting with line 248
  4. line 220: promising tool in "de" PPS care should be corrected 
  5. There seems to be a discrepancy in the total number of adverse events.  
    1. Table 9= 108
    2. Table 10=97
    3. Table 11=103
    4. Table 12=103
    5. Tale 13=103

Discussion

  1. The beginning of the second paragraph would be best served in the methods section to show how patients are chosen or excluded
  2. The authors should speak to the safety and the need for not only appropriate patient but also procedure selection as it seems that after excluding bronchoscopy and endoscopy to anesthesia resulted in significant reductions in adverse events. Inadequate sedation could be considered an adverse event if the child experiences trauma from the experience. 

Author Response

Reviewer 2

The authors retrospectively look at their Pediatric Procedural Sedation and Analgesia program with an eye towards safety and efficacy. I applaud the authors on taking this important step in program evaluation to continuously improve the program. The manuscript would benefit from consideration of the points listed below. 

Introduction: overall written colloquially. It could benefit from a review

Materials and Methods

  1. Authors should better define how various methods were chosen by the PROSA team. Where there guidelines or were they "prescribed" by the anesthesiologist overseeing the team?

Methods were chosen in consultation with the patient, parents and the prosa team, including the anesthesiologist. This has been added to the revised version. 

  1. Line 112: a couple of years is vague. The authors should better define the criteria for experience 

the criteria of experience is they need to have at least 3 years of experience as a nurse, not necessarily pediatric nurse, but having experience with children in a hospital setting like the PICU, ER or OR. This has been added to the revised version. 

  1. Line 117: why was only intermediate pediatric life support required? The authors reference in line 345 that a survey found that only a third of sites were advanced life support certified which assumes that this is the standard

There is a backup of a resuscitation team, so that intermediate pediatric life support training was felt to be sufficient in combination with confirmed back-up (cf telephone to inform anesthesiologist). The requirement is IPLS “Immediate Pediatric Life support” , which means that we as Prosa are the first responders when problems are rising and also the bridge to anesthesia if needed. The IPLS in combination with the yearly training in the operating room, a particular restrictive way of choice of patients (ASA 1 and ASA 2) and the confirmed backup by anesthesia has been approved by the anesthesia group. Additional information has been added to the revised version.

Results:

  1. For those MRI scans done in the context of the COSMO project, was this administered and trained by the PROSA team? If not, this may not represent an accurate assessment of the PROSA program for MRI. 

Only the MRI scans of the COSMO-project done and supported by a Prosa-nurse have been registered. We trained those patients and we didn’t administered anything because this procedure is done in a non-pharmacologic way. Those MRI’s wouldn’t have happened without the Prosa-approach. So YES it represents the Prosa-program. We have added the ‘non-pharm’ concept to the revised version of the paper.

  1. Table 5: is there a difference between MedVac Mattress and Mattress alone? The first line on many of the tables are underlined and it is unclear why or if there is an intentional separation 

No; We have adapted the wording to made this clearer, and have also adapted the table lines when appropriate.

  1. Table 5: Which medications were administered using those administrative routes? For example some medications have multiple routes of administration. There is a brief but incomplete expiation in the paragraph starting with line 248

As prosa-nurse we are only allowed to give certain medication through specific administrative routes. Let me visualize which medication we give in what administrative route:

  • Midazolam was, until gastroscopy and bronchoscopy was stopped, mostly given IV. Now the majority of Midazolam is given through a kind of sirop (PO) This is for anxious kids where for example we need to have a blood sample (combined with topical anesthesia).
  • Pethidine : was the combo medication for gastroscopy and bronchoscopy, so also only IV-route. The frequency of administration of this product has decreased firmly.
  • Fentanyl : only allowed to give by intranasal route
  • Dexdor : only allowed to give by intranasal route
  • Chloral hydrate : only oral or through GT

             We have added additional information to the revised version.

  1. line 220: promising tool in "de" PPS care should be corrected 

Done

  1. There seems to be a discrepancy in the total number of adverse events.  

    1. Table 9= 108
    2. Table 10=97
    3. Table 11=103
    4. Table 12=103
    5. Tale 13=103

We have tried to explain the differences, and have added this information in table 9 and 10, as the number of events is not equal to the number of cases with an AE (one case can have different AEs), and the same holds true when analysed based on the type and number of procedures.

Discussion

  1. The beginning of the second paragraph would be best served in the methods section to show how patients are chosen or excluded

Done

  1. The authors should speak to the safety and the need for not only appropriate patient but also procedure selection as it seems that after excluding bronchoscopy and endoscopy to anesthesia resulted in significant reductions in adverse events. Inadequate sedation could be considered an adverse event if the child experiences trauma from the experience. 

Totally agree that inadequate sedation must be considered as an adverse event, especially if the patient has a traumatic experience. This is why it has been classified as a failure. We are now also changing our registration chart so that can register not only if the sedation succeeded or failed, but also if the patient was comfortable (= not traumatized). We keep the question in the back of our head : will I proceed in the same way next time this patient needs sedation? Will the patient react the same way the next time we sedate him/her? A very important reflection which is very much appreciated by the parents. We refer to the reply to the first reviewer, as we have adapted our practices in the meanwhile with a shift to general anesthesia for broncho- and endoscopy. This has been adapted in the revised version of the paper.